# Enhancing Transcriptomic Insights into Neurological Disorders Through the Comparative Analysis of Shapley Values

José A. Castro-Martínez [1,†], Eva Vargas [1,†] , Leticia Díaz-Beltrán [1,2,*] and Francisco J. Esteban [1,*]

1 Systems Biology Unit, Department of Experimental Biology, Faculty of Experimental Sciences, University of Jaén, 23071 Jaén, Spain; j.a.castrobiologo@gmail.com (J.A.C.-M.); evargas@ujaen.es (E.V.)
2 Clinical Research Unit, Department of Medical Oncology, University Hospital of Jaén, 23007 Jaén, Spain
* Correspondence: ldiaz@fibao.es (L.D.-B.); festeban@ujaen.es (F.J.E.);
  Tel.: +34-953213630 (L.D.-B.); +34-953220306 (F.J.E.)
† These authors contributed equally to this work.

**Abstract:** Neurological disorders such as Autism Spectrum Disorder (ASD), Schizophrenia (SCH), Bipolar Disorder (BD), and Major Depressive Disorder (MDD) affect millions of people worldwide, yet their molecular mechanisms remain poorly understood. This study describes the application of the Comparative Analysis of Shapley values (CASh) to transcriptomic data from nine datasets associated with these complex disorders, demonstrating its effectiveness in identifying differentially expressed genes (DEGs). CASh, which combines Game Theory with Bootstrap resampling, offers a robust alternative to traditional statistical methods by assessing the contribution of each gene in the broader context of the complete dataset. Unlike conventional approaches, CASh is highly effective at detecting subtle but meaningful molecular patterns that are often missed. These findings highlight the potential of CASh to enhance the precision of transcriptomic analysis, providing a deeper understanding of the molecular mechanisms underlying these disorders and establishing a solid basis to improve diagnostic techniques and developing more targeted therapeutic interventions.

**Keywords:** autism; bipolar disorder; major depressive disorder; microarrays; schizophrenia; systems biology; transcriptomics

## 1. Introduction

Complex disorders affecting neurological processes are responsible for great health, social and economic costs worldwide. Despite the heterogeneity of these complex disorders, they all pose a significant global burden, since the misunderstanding of their causes and the associated factors that intensify the importance of these phenotypes is the main cause of the insufficiency of diagnosis; also, the lack of effectiveness in medical treatment for patients negatively impacts the well-being of those affected.

Autism Spectrum Disorder (ASD) [1,2] is a phenotype that spans the most severe autism, when social and communicative functions are very limited, to Asperger syndrome, characterized by mild symptoms. In any case, all diagnostic features show a rigid behavior and a pathological selection for some issues, and the capacity for attention and communication is affected [3]. Some body systems are also affected, such as digestive [4,5], immune [6–9], circulatory [10,11] and nervous [12–14]. Microbiota [15] and genetic causes [16] have been proposed in the early development of this disorder, and studies support the idea that the risk of suffering ASD rises when relatives are affected [17]. These symptoms are harmful for patients' autonomy and the welfare of caregivers [18,19]. The World Health Organization communicated in 2023 that one in 100 children is suffering from this disorder and its prevalence has been rising in the previous few years. Due to that, and given the fact that the origins and development of this condition are not agreed upon by specialists and researchers in this field of medicine [20,21], plenty of research teams are thinking of strategies to discover the etiology and main factors for understanding this disorder.

Schizophrenia is a neurological disorder that is characterized by positive (hallucinations, lack of social skills and cognitive distortions) and negative (general apathy, social and job issues) [22] symptoms. This disease is linked to increased vulnerability to cardiovascular [23,24], metabolic [25] and infectious [26] diseases, which raise the risk of an early death. Furthermore, it has a direct link with suicide index growth [27–29]. Also, caregivers and relatives are negatively affected in social terms, since patients suffer from diminished autonomy [30,31]. Its prevalence worldwide is 24 million people [32,33], with a percentage of 0.32–0.45% in adults [34]—and it tends to appear in teenagers at an advanced age [35]. To date, the origin of this disorder remains unknown [36]. On this note, there is some consensus in the relevance of some gene factors implied in its onset [25], but this is not determinant to its origin [37], considering that other factors such as social environment, drug abuse (including alcohol) [38–40], and neural pruning, usual in adolescence, can have a decisive influence [35]. Myelin sheaths [41] and central nervous system architecture [42–44] are also bonded with this disorder. The most extended belief nowadays is that schizophrenia is a multifactorial disorder [45,46]. In order to clarify the disease causes, omics techniques such as transcriptomics have been applied [47]. Nevertheless, this is a pathological situation that harms the life quality of patients in a severe way, generating a medical and social interest that concerns the pharma industry, which attempts to alleviate this suffering with drugs that minimize the secondary effects associated with available treatments [48], usually adverse for the daily life of patients [49]. Thus, efficient research is crucial to solve the social and economic problems attached to this disease [50].

Bipolar disorder (BD) [51,52] is a neurological condition characterized by the alternance of manic episodes (euphoria, excessive joy, uncontrolled enthusiasm, etc.) with depressive ones (anhedonia, sadness, lack of interest in living, etc.) [53]. Genetic causes have been studied [47], and some environmental factors such as alcoholism and other types of drug abuse have been proposed as a disease cause [54]. The development of genomics and transcriptomics may help us to understand the disorder and treat it efficiently. Its prevalence was 40 million people in 2019 [55], and being affected by BD raised the suicide index for these patients [56–58]. There is still not much understanding of this disorder, but some drugs, including lithium, have been reported to alleviate its symptoms [59,60].

Major depressive disorder (MDD) [61] is a neurologic disease of unknown origin [62], with more severe symptoms than common depression [63]. Among these are anhedonia, sadness and a lack of desire to live [64]. Genetic causes are considered, which has led to the development of transcriptomics and epigenetic studies [65]. Physiological and hormonal origins have also been reported, as well as environmental factors like stress and psychological and social aspects [66]. Due to the fact that its origin remains unknown, it is classified as a complex disorder [67], which causes a great social and economic burden for the community environment of the affected people [68,69]. World prevalence is about 350 million people [70,71], but there is not much consensus. In fact, this prevalence differs among regions (3% in Japan and 16.3% in USA) [70,71]. Every year as many as 850,000 suicides due to major depressive disorder have been registered [72,73]. Different techniques, such as omics and neuroimaging, and several biomarkers such as certain fatty acids and miRNA have been used, but there is no consensus [74–76]. Nowadays, there are lots of medicines that treat this disease, taking advantage of the limited knowledge we have about the brain.

Despite their high prevalence worldwide, the origins of these disorders are still unknown. Because of that, it is necessary to apply techniques that are able to detect key factors for prevention and treatment, pointing towards their main causes and improving the health and quality of life of these patients as much as possible.

Advances in omics technologies, specifically at the microarray analysis level, have revolutionized the thorough exploration of gene expression patterns linked to complex neurological phenotypes [77–79]. Microarray technology enables the simultaneous measurement of thousands of genes, providing deep insights into the altered molecular mechanisms implicated in the etiopathogenesis of various diseases [80–84]. A critical aspect of microar-

ray data analysis is the identification of differentially expressed genes (DEGs), which serve as key indicators in understanding disease mechanisms. Traditionally, these analyses have relied on ranking genes based on individual *p*-values; however, this approach does not always correlate with biological significance. In some cases, small *p*-values, indicative of high statistical significance, may not correspond to biologically relevant signals, while larger *p*-values, often disregarded, could be linked to genes crucial for specific biological processes [85]. Classical microarray analysis methods typically utilize Welch's *t*-test and linear models such as Empirical Bayes to identify DEGs by comparing gene expression levels between experimental groups or situations [86,87]. However, these traditional approaches may miss significant gene expression changes, particularly in complex diseases like those affecting the brain, which are characterized by heterogeneous molecular profiles [88,89].

To address the limitations of *p*-value-based methods, which may often result in a loss of biologically relevant information due to multiple testing correction techniques, more reliable methodologies have been implemented [85,90,91]. Remarkably, there is one statistical technique that applies Game Theory, utilizing a computational concept known as the Shapley value [85]. This approach offers a more refined evaluation of the significance of each gene by assessing the cumulative contribution of each transcript within the context of the whole gene set under analysis. The Shapley value measures the relevance of each gene by assessing its contribution alongside the contributions of the rest of the genes in the experiment [92]. By combining Game Theory with classical statistical analyses, this methodology provides a powerful tool to improve the detection and interpretation of relevant differences at the gene expression level [85].

We applied the microarray games methodology in this study, specifically harnessing Shapley values, to gene expression data from different neurological pathologies. This approach integrates Game Theory with the aim of improving the detection and functional analysis of genes involved in complex neurological conditions, such as ASD, schizophrenia, bipolar disorder, and major depressive disorder [85]. By evaluating the contribution of each transcript across all possible coalitions, this technique reveals powerful knowledge about the genetic underpinnings of these complex diseases, potentially leading to innovative diagnostic and therapeutic strategies. Comparative Analysis of the Shapley values approach not only enhances the identification of key molecular players but also enhances our understanding of their biological activities within complex, multi-genic contexts.

To achieve a comprehensive understanding of the gene expression profiles associated with four prevalent neurological pathologies, we employed two distinct methods for microarray data analysis: (i) an orthodox approach utilizing Welch's *t*-test and Empirical Bayes methods, and (ii) an alternative analysis based on the Comparative Analysis of Shapley values (CASh) method, derived from Game Theory. Previous research [85,93] has demonstrated that the CASh method significantly increases the power to detect differentially expressed genes (DEGs), providing a more robust framework for analyzing complex biological data.

## 2. Materials and Methods

### 2.1. Microarray Expression Data Retrieval, Processing and Exploratory Analysis

Microarray-derived gene expression data were sourced from the public repository Gene Expression Omnibus (GEO) https://www.ncbi.nlm.nih.gov/geo/ (accessed on 18 March 2024). For dataset selection, raw data from Affymetrix commercial microarrays were prioritized whenever available.

CEL files from three autism datasets—GSE6575, GSE18123, and GSE25507 [94–96]; two schizophrenia datasets—GSE17612 and GSE62333 [97,98]; two bipolar disorder datasets—GSE5389 and GSE7036 [99,100]; and two datasets encompassing schizophrenia, bipolar disorder, and major depressive disorder samples—GSE12654 and GSE53987 [101,102] were accessed through the GEO repository. Raw data were downloaded for each dataset, and further preprocessing, quality control, and normalization steps were applied using relative log expression (RLE), normalized unscaled standard error (NUSE), and Robust Multi-Array

Average expression measure (RMA) techniques. The 'affy' (version 1.82.0) and 'affyPLM' (version 1.80.0) RStudio packages were used for these parts of data analysis (RStudio version 2021.09.20) [103–105]. Finally, expression matrices were created, and samples were assigned to experimental and control groups for further analysis (Supplementary Table S1).

Each dataset was analyzed separately to identify differentially expressed genes (DEGs). Two distinct approaches were employed for differential expression analysis between patients and controls: (i) a traditional approach utilizing Welch's *t*-test and Empirical Bayes methods, and (ii) an alternative method based on the Comparative Analysis of Shapley values (CASh) technique.

To gain a comprehensive understanding of gene expression patterns, we applied various microarray data exploratory techniques. Principal Component Analysis (PCA), heatmaps, and volcano plots were generated to evaluate the distribution of gene expression patterns. PCA was used to illustrate gene expression distribution at two levels: (i) the entire gene set in each dataset and (ii) the DEGs identified by CASh analysis (*p*-value < 0.01). Heatmaps were created to visualize DEGs after differential gene expression analysis through Empirical Bayes (raw *p*-value < 0.05) and CASh (*p*-value < 0.01) methods, which exhibited a well-defined clustering of samples according to disease status. Additionally, volcano plots were employed to contrast the *p*-values obtained from Empirical Bayes and CASh analyses, providing a visual representation of the statistical relationships between the methods used for detecting DEGs (see Supplementary Figures S1–S3 for further details).

### 2.1.1. Traditional Approaches

Orthodox analyses for detecting DEGs were conducted using the unequal variance *t*-test (Welch's *t*-test), implemented through the 'multtest' (2.60.0) package in RStudio (version 2021.09.0) [106]. In microarray experiments, the small number of replicates and the large number of genes typically analyzed pose significant challenges, leading to the issue of low statistical power with ordinary *t*-tests. This limitation makes *t*-tests less effective for filtering out regulated genes [107,108]. Moreover, most multiple testing adjustments tend to be quite conservative, particularly with small replicate numbers [108]. To address this issue, we employed Bayesian-based methods, specifically the Empirical Bayes approach, as implemented in the Bioconductor 'limma' R package (version 4.4, https://bioconductor.org/packages/release/bioc/html/limma.html, accessed on 22 October 2024).

Significant DEGs were identified after multiple testing correction with the Benjamini and Hochberg method to control the False Discovery Rates (FDR) [109].

### 2.1.2. Comparative Analysis of Shapley Values (CASh) Approach

We utilized the Comparative Analysis of Shapley values (CASh) method to the detection of differentially expressed transcripts by assessing their cooperative contribution to overall changes at gene expression levels [110]. The Shapley value, a concept rooted in Game Theory, quantifies the particular contribution of each gene to the overall expression changes observed in the whole dataset [111]. CASh combines the Microarray Game algorithm, which is applied to transcriptomic data from microarrays, with the Bootstrap technique that involves the resampling of some values to reduce the impact of potential outliers in the data matrix [110–113]. In this approach, gene expression is treated as a cooperative game, where each gene contributing collaboratively to the expression changes is detected, providing a more nuanced understanding of gene interactions within the dataset. The protocol used has been previously described in the articles by Esteban and Wall, 2011 and Castro-Martínez et al., 2024, where the mathematical development can also be found [85,93].

In our study, the CASh method was used for the detection of DEGs using two levels of stringency by setting cutoff *p*-values at 0.01 (more restrictive) and 0.05 (less restrictive). These genes were then analyzed to distinguish dysregulated expression levels (either over- or under-expression) in the experimental samples compared to the control group. Boolean

matrices were created to determine the expression levels, which were subsequently used to define microarray games and calculate the Shapley values.

A final matrix with the expression levels of a selected number of samples and genes was generated from the original data. The matrix included genes with raw *p*-values below 0.01 or 0.05 and classified the samples into experimental groups (e.g., patients with specific conditions) and healthy controls. To detect over-regulated gene expression levels relative to controls, each value in the gene expression vector was coded as 1 if it met or exceeded the mean plus the standard deviation of the control group expressions, and as 0 otherwise. This generated a Boolean matrix {0, 1} reflecting these criteria.

A similar method was employed to identify under-regulated expression: values below the mean minus one standard deviation of the control group was coded as 1, while all other values were coded as 0. This resulted in another Boolean matrix, where rows corresponded to genes and columns to samples. These Boolean matrices were then grouped by sample categories, creating separate matrices for each group. Using these group-specific matrices, microarray games were constructed for each condition, and Shapley values were computed to evaluate the significance of each gene's contribution to the conditions under study.

To attenuate the influence of random high Shapley values, a Bootstrap resampling procedure was applied with 1000 iterations for each analysis, similar to that described by Moretti et al. (2008) [110]. This method, known as Comparative Analysis of Shapley values (CASh), refines the detection of genes significantly associated with the studied conditions.

To further reduce the likelihood of false positives, multiple testing corrections were applied, and Shapley values were compared against statistically significant thresholds. Additionally, Fold Changes (FC) were evaluated, with genes exhibiting *p*-values below 0.01 and 0.05 and $|FC| > 2$ being considered statistically significant.

### 2.2. Biological Pathway Analysis and Functional Profiling

The g:Profiler functional profiling tool (version e111_eg58_p18_30541362), specifically the g:GOSt module https://biit.cs.ut.ee/gprofiler/gost (accessed on 10 September 2024), was utilized to perform a functional enrichment analysis of the biological processes and pathways influenced by differentially expressed genes (DEGs). This tool leverages Gene Ontology (GO) terms to create a comprehensive overview of gene functions and interactions [114,115]. Gene Ontology provides a structured vocabulary that classifies and integrates biological data across species into three main categories: biological processes (BPs), cellular components (CCs), and molecular functions (MFs).

During the analysis of gene expression data such as those derived from the application of microarray devices, it is crucial to ensure that gene identifiers (IDs) are accurately annotated and standardized to the official gene symbols, especially when consolidating data from different sources. The aim of this step is to further facilitate meaningful biological interpretation, which ultimately enhances the consistency and reliability of genomic data analysis, The g:Convert tool, available on the g:Profiler web server https://biit.cs.ut.ee/gprofiler/convert (accessed on 10 June 2024, was used for this purpose. The g:Convert module supports a plethora of biological identifiers, including Ensembl IDs, UniProt IDs, RefSeq, and others, allowing researchers to input data from various experimental outputs and databases. In cases of ambiguity of the transcript names, which can occur due to multiple identifiers for a single gene or updates in genomic databases, we prioritized IDs with the most extensive Gene Ontology (GO) annotations. By selecting IDs with the most GO annotations, we aimed to enhance the robustness of our dataset, ensuring that the functional analysis reflects supported gene functions and interactions well [114,115].

To determine the significance of the GO categories analyzed, we applied a rigorous statistical criterion, the Benjamini-Hochberg False Discovery Rate (FDR), with GO terms with an FDR value below 0.05 considered as significantly enriched, thus minimizing the likelihood of false positives. To visually represent the findings, the significantly enriched GO terms in each category were plotted for CASh 0.05 comparisons using the 'ggplot2' (version 3.5.1) package in RStudio [116].

## 3. Results

### 3.1. Data Collections and Samples Analyzed

Microarray expression data from nine datasets, encompassing 506 samples in total, were included in this study. Table 1 provides an overview of the main characteristics of these datasets.

**Table 1.** Gene Expression Omnibus (GEO) datasets accessed in our study. For each study, the main characteristics of samples are shown. SCH: Schizophrenia; BD: Bipolar Disorder; MDD: Major Depressive Disorder.

| Phenotype Group | Dataset ID | No. of Samples | Type of Samples |
|---|---|---|---|
| Autism | GSE6575 | 25 | Whole blood autism (n = 14) vs. controls (n = 11) |
| | GSE18123 | 23 | Whole blood autism (n = 13) vs. controls (n = 10) |
| | GSE25507 | 26 | Peripheral blood lymphocytes (n = 12) vs. controls (n = 14) |
| Schizophrenia | GSE17612 | 30 | Brain tissue (n = 17) vs. controls (n = 13) |
| | GSE62333 | 25 | Skin fibroblasts (n = 11) vs. controls (n = 14) |
| Bipolar disorder | GSE5389 | 17 | Brain tissue (n = 7) vs. controls (n = 10) |
| | GSE7036 | 6 | Lymphoblastoid cell lines (n = 3) vs. controls (n = 3) |
| Miscellanea (SCH, BD, MDD) | GSE12654 | 38 | Brain tissue (n = 24) vs. controls (n = 14) |
| | GSE53987 | 186 | Brain tissue (n = 135) vs. controls (n = 51) |

To detect differentially expressed genes (DEGs), we employed two distinct strategies. First, traditional methods based on the unequal variance *t*-test (Welch's *t*-test) and Empirical Bayes were applied. Following this, we conducted an alternative analysis using the CASh method. The conventional approach, utilizing Welch's *t*-test and Empirical Bayes, failed to identify any DEGs. In contrast, the CASh method successfully revealed several transcripts when using both 0.01 and 0.05 cutoff *p*-values for the preselection of DEGs (Table 2). Complete lists of DEGs detected for each dataset after these comparisons are provided in Supplementary Table S2. Our analyses demonstrate that the use of the CASh method significantly improves the detection of DEGs across the nine datasets analyzed.

**Table 2.** Number of differentially expressed genes (DEGs) detected after the differential gene expression analysis using conventional techniques (unequal variances Welch's *t*-test and Empirical Bayes (EBayes) [117]), and alternative approaches based on the Comparative Analysis of Shapley values (CASh) method with raw *p*-values thresholds of 0.01 or 0.05, respectively. FDR corrected *p*-values are calculated where indicated. SCH: Schizophrenia; BD: Bipolar Disorder; MDD: Major Depressive Disorder; HPC: hippocampus; PFC: pre-frontal cortex; STR: striatum.

| Dataset ID | Welch's *t*-Test | EBayes FDR < 0.01 | EBayes FDR < 0.05 | CASh 0.05 FDR < 0.05 | CASh 0.01 | CASh 0.05 |
|---|---|---|---|---|---|---|
| GSE6575 | 0 | 0 | 0 | 0 | 204 (87 ↑, 117 ↓) | 930 (324 ↑, 606 ↓) |
| GSE18123 | 947 | 205 | 2973 | 45 (12 ↑, 33 ↓) | 879 (467 ↑, 412 ↓) | 1862 (1027 ↑, 835 ↓) |
| GSE25507 | 0 | 0 | 0 | 0 | 28 (10 ↑, 18 ↓) | 141 (41 ↑, 100 ↓) |
| GSE17612 | 0 | 0 | 0 | 0 | 1 (1 ↑, 0 ↓) | 11 (8 ↑, 3 ↓) |
| GSE62333 | 5 | 0 | 5 | 0 | 68 (33 ↑, 35 ↓) | 164 (95 ↑, 69 ↓) |
| GSE5389 | 1 | 0 | 2 | 0 | 40 (24 ↑, 16 ↓) | 162 (103 ↑, 59 ↓) |
| GSE7036 | 0 | 0 | 0 | 0 | 8 (4 ↑, 4 ↓) | 35 (12 ↑, 23 ↓) |
| GSE12654_SCH | 0 | 0 | 0 | 0 | 2 (2 ↑, 0 ↓) | 8 (4 ↑, 4 ↓) |
| GSE12654_BD | 0 | 0 | 0 | 0 | 0 | 8 (6 ↑, 2 ↓) |
| GSE12654_MDD | 0 | 0 | 0 | 0 | 0 | 0 |
| GSE53987_HPC_SCH | 283 | 2 | 1393 | 4 (0 ↑, 4 ↓) | 794 (595 ↑, 199 ↓) | 655 (357 ↑, 298 ↓) |
| GSE53987_HPC_BD | 0 | 0 | 0 | 0 | 41 (14 ↑, 27 ↓) | 152 (48 ↑, 104 ↓) |
| GSE53987_HPC_MDD | 0 | 0 | 0 | 0 | 47 (14 ↑, 33 ↓) | 163 (41 ↑, 122 ↓) |

**Table 2.** *Cont.*

| Dataset ID | Welch's t-Test | EBayes FDR < 0.01 | EBayes FDR < 0.05 | CASh 0.05 FDR < 0.05 | CASh 0.01 | CASh 0.05 |
|---|---|---|---|---|---|---|
| GSE53987_PFC_SCH | 0 | 0 | 32 | 0 | 182 (106 ↑, 76 ↓) | 354 (179 ↑, 175 ↓) |
| GSE53987_PFC_BD | 0 | 0 | 0 | 0 | 157 (54 ↑, 103 ↓) | 394 (141 ↑, 253 ↓) |
| GSE53987_PFC_MDD | 0 | 0 | 0 | 0 | 61 (11 ↑, 50 ↓) | 175 (34 ↑, 141 ↓) |
| GSE53987_STR_SCH | 0 | 0 | 1 | 0 | 81 (36 ↑, 45 ↓) | 258 (139 ↑, 119 ↓) |
| GSE53987_STR_BD | 0 | 0 | 0 | 0 | 42 (10 ↑, 32 ↓) | 77 (33 ↑, 44 ↓) |
| GSE53987_STR_MDD | 0 | 0 | 0 | 0 | 19 (8 ↑, 11 ↓) | 32 (12 ↑, 30 ↓) |

Among the main DEGs detected through the application of CASh, *PCDC4* (Programmed Cell Death 4), *BRF1* (BRF1 RNA Polymerase III Transcription Initiation Factor Subunit), *OFCC1* (Orofacial Cleft 1 Candidate 1 (Pseudogene)), *TMTC1* (Transmembrane O-Mannosyltransferase Targeting Cadherins 1), *TIPRL* (TOR Signaling Pathway Regulator), *LEPROT* (Leptin Receptor Overlapping Transcript), *NR5A2* (Nuclear receptor subfamily 5 group A member 2), and *ATM* (Ataxia Telangiectasia Mutated Serine/Threonine Kinase) are some examples of frequently dysregulated genes detected in the datasets of Autism Spectrum Disorder analyzed in our study.

*GADD45B* (Growth Arrest And DNA Damage Inducible Beta), *UTP4* (UTP4 small subunit processome component) and *TNFRSF10A* (TNF receptor superfamily member 10a) were common DEGs to three of the schizophrenia datasets analyzed, and *PDHA1* (Pyruvate Dehydrogenase E1 Subunit Alpha 1), *CCDC91* (Coiled-Coil Domain Containing 91), *CHD9* (Chromodomain Helicase DNA Binding Protein 9), and *SMIM14* (Small Integral Membrane Protein 14) were DEGs in at least two out of the six schizophrenia datasets included in our study.

*VWA8* (von Willebrand domain-containing protein 8), *SNAP29* (Synaptosomal-associated protein 29), *RIF1* (Replication Timing Regulatory Factor 1), *AQP4* (Aquaporin-4) and *GSTM3* (Glutathione S-Transferase Mu 3) were some relevant differentially expressed genes detected in the bipolar disorder datasets.

Finally, the variety of DEGs detected in the Major Depressive Disorder datasets was high, and only the gene *CCDC144A* (Coiled-Coil Domain Containing 144A) was dysregulated in more than one of the datasets analyzed. Nevertheless, some other relevant overexpressed genes detected after CASh analysis of the major depression datasets include *EXOSC2* (Exosome Component 2), *DPP10* (Dipeptidyl Peptidase Like 10), *GSTM5* (Glutathione S-Transferase Mu 5), and *ZNF184* (Zinc Finger Protein 184).

Regarding the main differentially expressed genes detected, it is also noteworthy to mention that some overlap was detected among different disorders. Bipolar disorder, schizophrenia and major depression shared two DEGs: *PRDX6* (Peroxiredoxin 6) and *GHRHR* (Growth-hormone-releasing hormone receptor). Autism Spectrum Disorder, schizophrenia and major depression exhibited an overlap of two DEGs as well: *SLC4A4* (Solute carrier family 4 member 4) and *Y_RNA* (RNA Gene Y RNA). Also, we were able to detect an overlap of six DEGs among the autism spectrum disorder, schizophrenia and bipolar disorder datasets: *RAB2A* (Member RAS Oncogene Family), *RAD23B* (RAD23 homolog B, nucleotide excision repair protein), *LGALS8* (Galectin-8), *PIAS1* (Protein Inhibitor of Activated STAT 1), *PDP1* (Pyruvate Dehydrogenase Phosphatase Catalytic Subunit 1), and *CHD9* (Chromodomain Helicase DNA Binding Protein 9). This overlap may indicate the existence of a common dysregulation at the gene expression level that could lead to the development of the neuropsychiatric conditions under study. However, we were not able to find any common DEG among the four neurological disorders analyzed. Thus, further research is warranted to help unravel the molecular origin of these complex neuropsychiatric disorders.

### 3.2. Functional Annotation Analysis of the Differentially Expressed Genes

The number of DEGs detected after the CASh 0.01 method and the FDR correction of *p*-values was insufficient to identify significantly enriched biological pathways associated with some gene sets. However, gene set enrichment analysis using the DEGs identified with the CASh 0.05 method revealed several significantly enriched processes in most of the analyzed datasets.

In the Autism Spectrum Disorder datasets (GSE6575, GSE18123, and GSE25507), DEGs were primarily associated with biological processes (BP) such as structure development, transport and cardiac development, while cellular components (CC) were mainly related to the cytoplasm and the intracellular organelles, and the molecular functions (MFs) were mainly associated with protein binding (Figure 1).

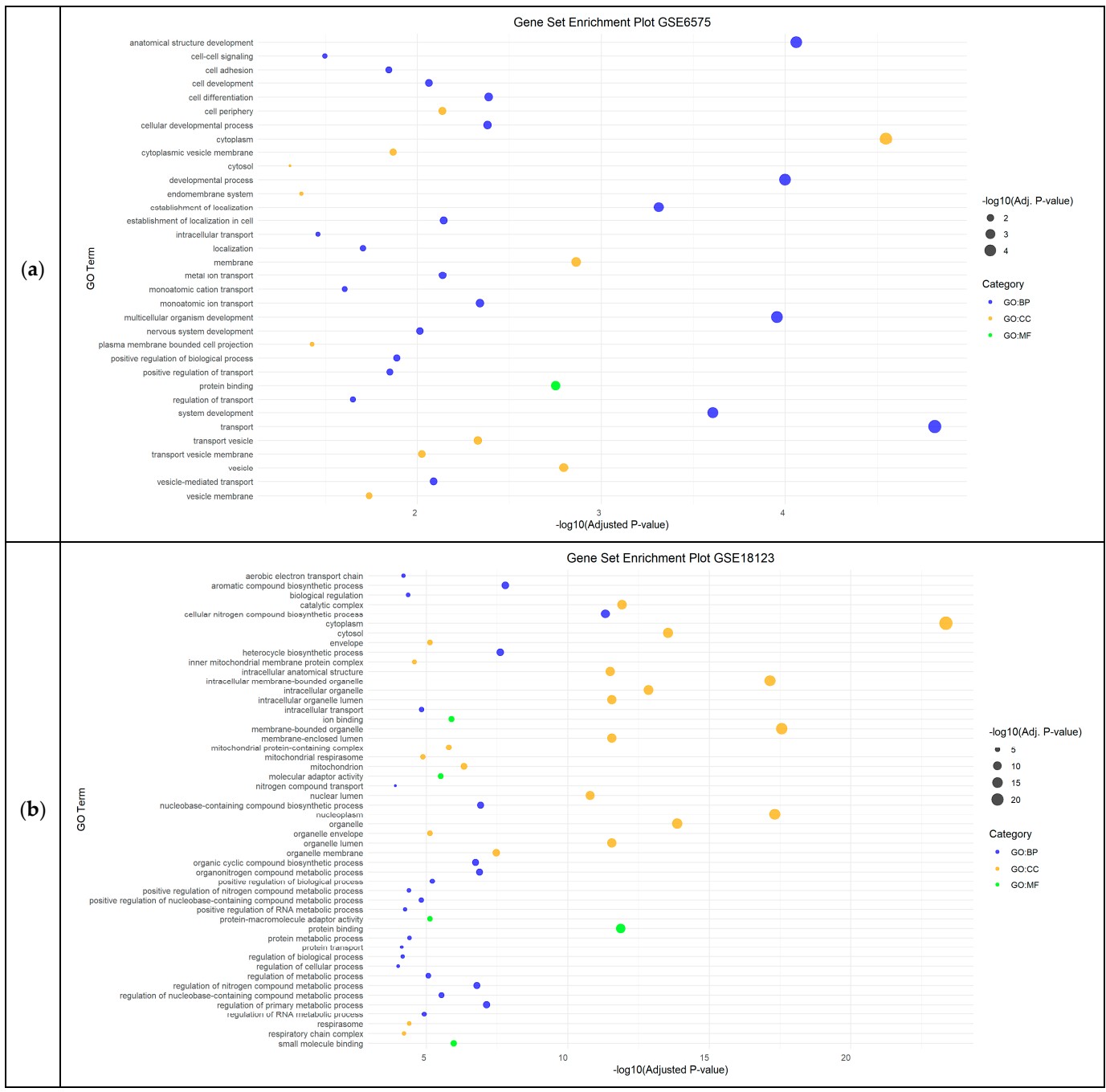

**Figure 1.** *Cont.*

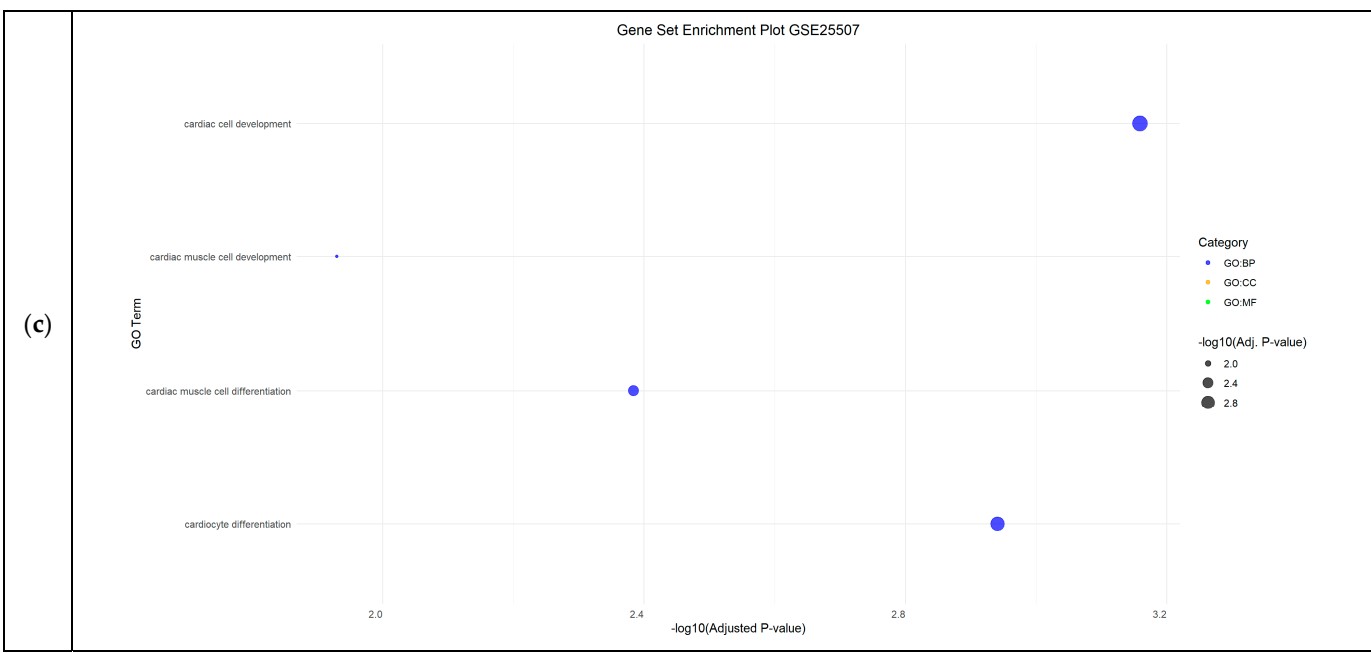

**Figure 1.** Functional Enrichment Analysis results showing significant Gene Ontology (GO) terms of the differentially expressed genes in Autism Spectrum Disorder datasets: (**a**) relevant GO terms identified in GSE6575 dataset; (**b**) top 50 GO terms in GSE18123 dataset; (**c**) GO significant terms in GSE25507 dataset. For each dataset, significantly enriched molecular functions (GO:MF), biological processes (GO:BP) and cellular components (GO:CC) are shown in green, blue and orange, respectively.

For the schizophrenia datasets (GSE17612, GSE62333, GSE12654, and GSE53987), the most significantly enriched BPs were related to the regulation of programmed cell death, regulation of primary metabolic processes, and the development of multicellular organism structures. The CC results highlighted cytoplasm, nucleoplasm, and extracellular space, while molecular function (MF) analysis identified activities mainly associated with glycine-tRNA ligase and protein binding activity (Figure 2).

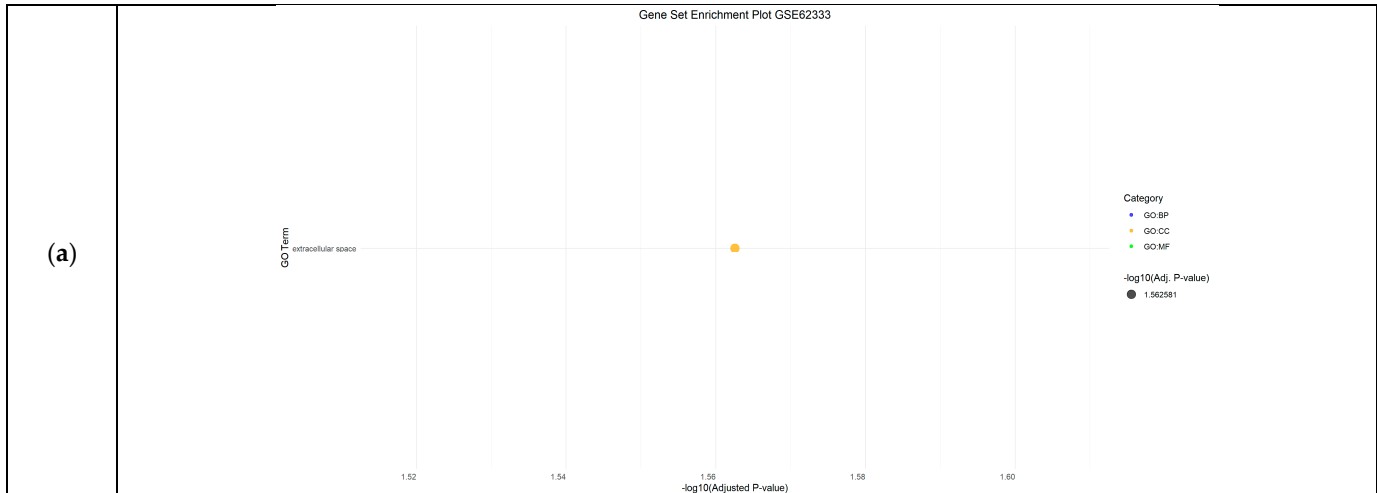

**Figure 2.** *Cont.*

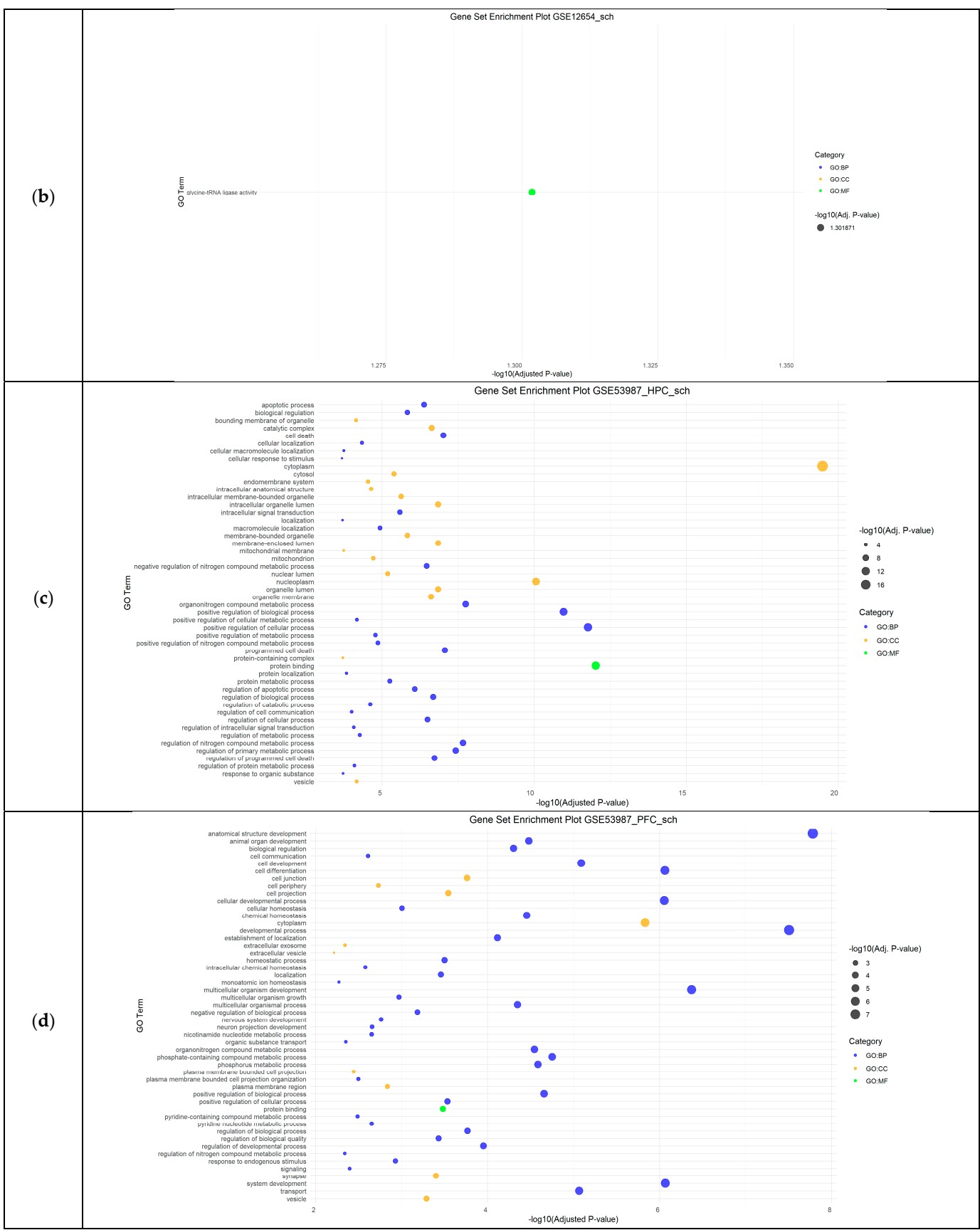

**Figure 2.** *Cont.*

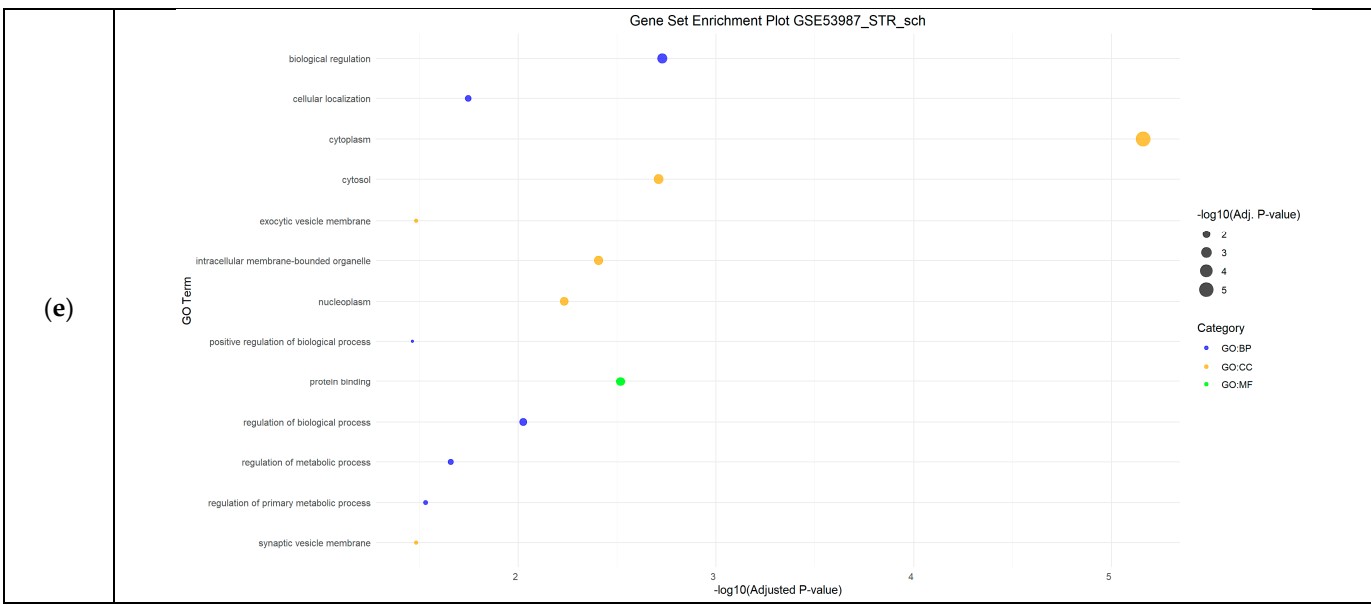

**Figure 2.** Functional Enrichment Analysis results showing significant Gene Ontology (GO) terms of the differentially expressed genes in Schizophrenia datasets: (**a**) GSE62333 dataset; (**b**) GSE12654_SCH dataset; (**c**) top 50 GO terms detected in GSE53987_HPC_sch dataset; (**d**) top 50 significant GO terms in GSE53987_PFC_sch dataset; (**e**) GSE53987_STR_sch dataset. For each dataset, significantly enriched molecular functions (GO:MF), biological processes (GO:BP) and cellular components (GO:CC) are shown in green, blue and orange, respectively. GSE17612 has no results in GO. SCH: schizophrenia; HPC: hippocampus; PFC: pre-frontal cortex; STR: striatum.

In the bipolar disorder datasets (GSE5389, GSE7036, GSE12654, and GSE53987), gene set enrichment analysis revealed neurogenesis and telomeric and metabolic processes (mainly those related to lipids and nitrogen compounds) as a significantly enriched BP. Additionally, synapses, lipoprotein activity and chromosomes, and protein binding were identified as significantly enriched CC and MF, respectively (Figure 3).

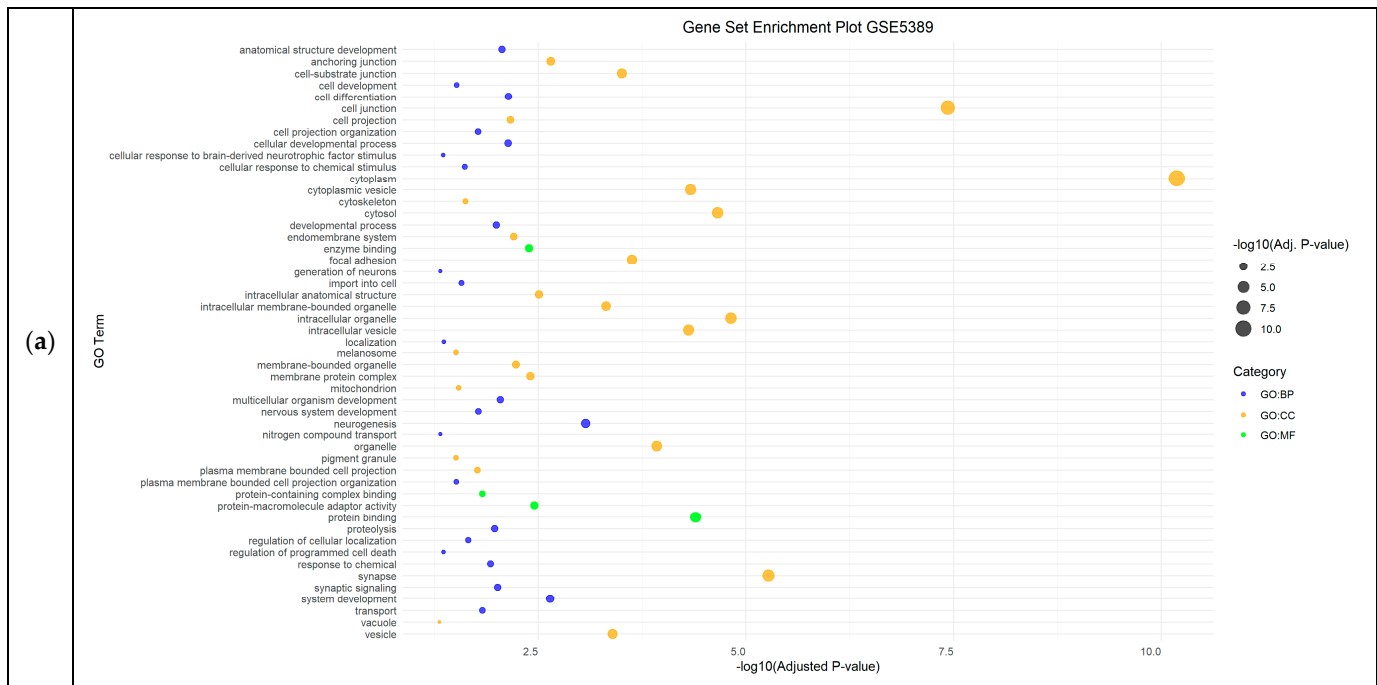

**Figure 3.** *Cont.*

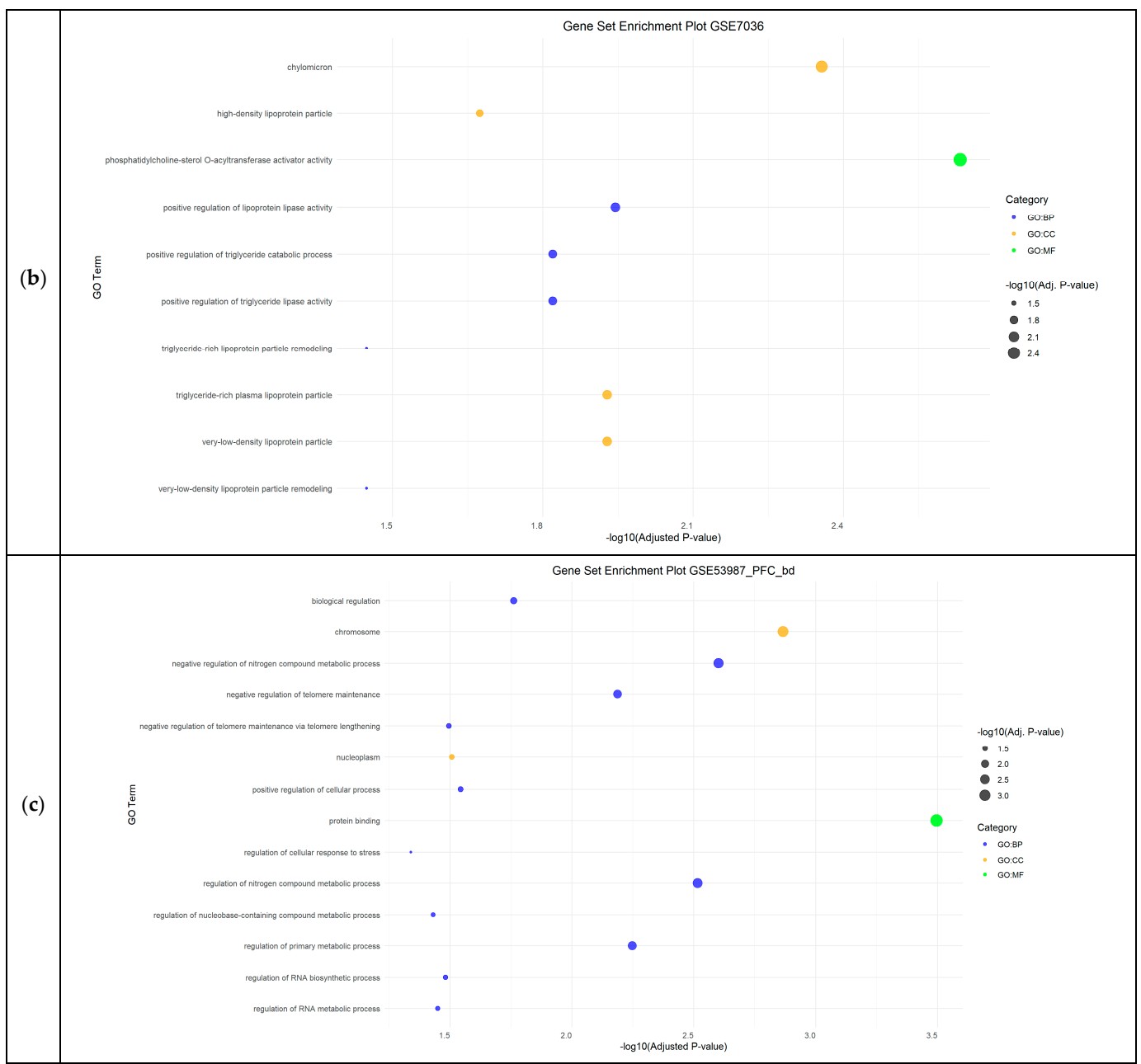

**Figure 3.** Functional Enrichment Analysis results showing significant Gene Ontology (GO) terms of the differentially expressed genes in Bipolar Disorder datasets: (**a**) GSE5389 dataset; (**b**) GSE7036 dataset; (**c**) GSE53987_PFC_bd dataset. For each dataset, significantly enriched molecular functions (GO:MF), biological processes (GO:BP) and cellular components (GO:CC) are shown in green, blue and orange, respectively. GSE12654, GSE53987_HPC_bd and GSE53987_STR_bd have no results in GO. BD: bipolar disorder; HPC: hippocampus; PFC: pre-frontal cortex; STR: striatum.

Finally, in the major depressive disorder datasets (GSE12654 and GSE53987), gene set enrichment analysis highlighted epiboly and wound healing as significantly enriched BPs, while nucleoplasm, the site of polarized growth, and the growth cone were identified as significantly enriched CCs and protein binding as significantly enriched MF (Figure 4).

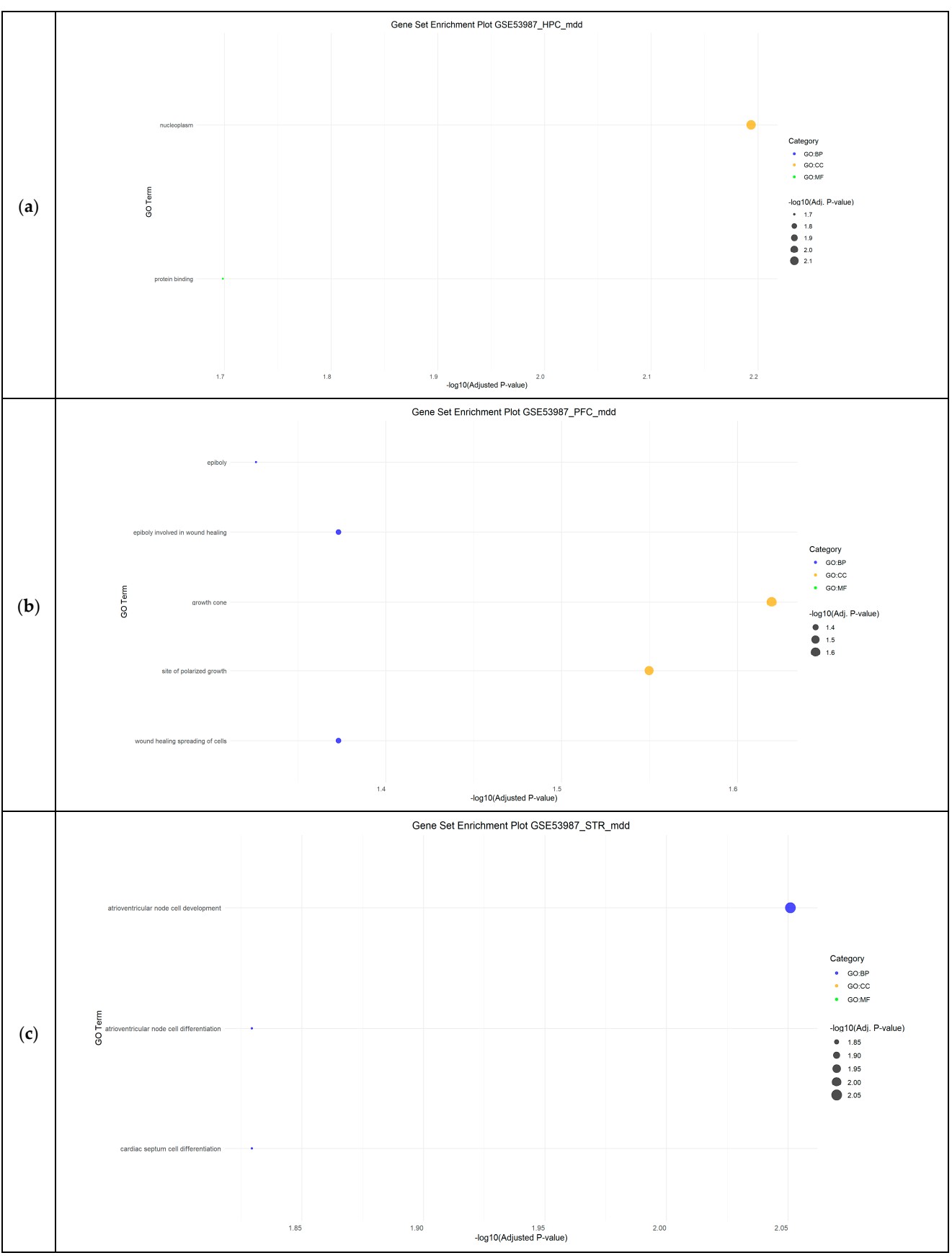

**Figure 4.** Functional Enrichment Analysis results showing significant Gene Ontology (GO) terms of the differentially expressed genes in Major Depression datasets: (**a**) GSE53987_HPC_mdd dataset;

(**b**) GSE53987_PFC_mdd dataset; (**c**) GSE53987_STR_mdd dataset. For each dataset, significantly enriched molecular functions (GO:MF), biological processes (GO:BP) and cellular components (GO:CC) are shown in green, blue and orange, respectively. GSE12654 has no results in GO. MDD: major depressive disorder; HPC: hippocampus; PFC: pre-frontal cortex; STR: striatum.

At this point, it is important to highlight the overlap detected between the GO terms identified after the gene set enrichment analysis (Figure 5). Notably, the molecular function "protein binding" (GO:0005515) and the cellular component "nucleoplasm" (GO:0005654) were revealed as significant GO terms in all the neuropsychiatric conditions analyzed.

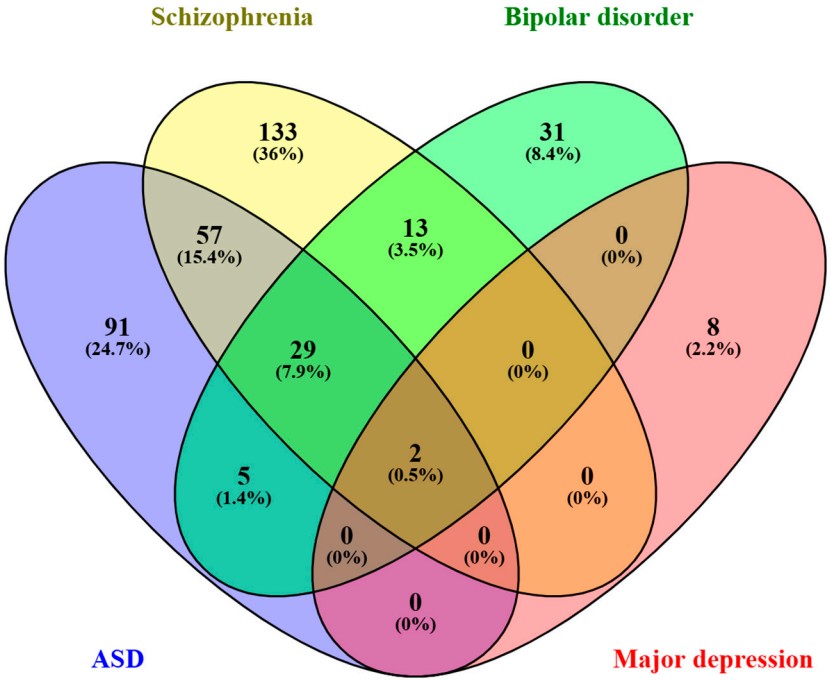

**Figure 5.** Venn diagram showing significant Gene Ontology (GO) terms associated with the four neuropsychiatric conditions analyzed. ASD: Autism Spectrum Disorder. The diagram was built using Venny's online tool available at: https://bioinfogp.cnb.csic.es/tools/venny/ (Oliveros, 2015) (accessed on 22 October 2024).

## 4. Discussion

Neurological pathologies inflict significant suffering and pose substantial burdens on millions of people worldwide. Lately, a significant boost to omics technologies has enabled a comprehensive exploration of the molecular mechanisms underlying a number of common neurological conditions. Microarrays, which emerged nearly three decades ago aiming to study whole gene expression profiles, have since shed light on the molecular pathways involved in disease pathogenesis [118,119]. In the present work, we investigated data from nine public datasets obtained using Affymetrix microarray devices, including three datasets from Autism Spectrum Disorder, two from Schizophrenia, two from Bipolar Disorder, and two datasets encompassing samples from Schizophrenia, Bipolar Disorder, and Major Depressive Disorder.

Raw data were accessed through the GEO public repository, and gene expression files were pre-processed, quality controlled, and normalized. Then, to the detection of differentially expressed genes (DEGs), we employed two strategies: (i) a traditional approach using classical unequal variances *t*-test and Empirical Bayes methods, and (ii) an alternative approach utilizing the CASh method [110]. The traditional *t*-test approach identified few DEGs, whereas the CASh method revealed a significant number of statistically relevant genes across the nine datasets analyzed. The *t*-test identifies genes based on their

differential expression between two conditions, considering a gene significant when its *p*-value falls below a pre-established threshold (0.05 adjusted *p*-value and |FC| > 2 in our study). In contrast, the CASh method not only considers the expression of each gene under two conditions but also evaluates the contribution of each gene across all possible permutations using the Shapley value as a measure. This holistic approach mitigates the impact of confounding variables by considering the global gene network rather than isolated gene expressions. However, a current limitation of the CASh method is that it does not explicitly account for potential confounding effects, which should be addressed in future applications [110,112,113,120]. In summary, CASh offers a more nuanced understanding of gene interactions and their collective impact on disease pathophysiology.

The use of CASh evidenced the dysregulation of genes previously known in the context of some of the conditions analyzed. Notably, risk variants in the *OFCC1* gene, down-regulated in some of the analyzed Autism Spectrum Disorder datasets, have been previously linked to ASD and other neurobehavioral disorders such as Tourette syndrome [121]. It has also been suggested that the gene *ATM* (up-regulated in our study) shapes the development of the GABAergic system, and its abnormal expression may affect the autistic condition in animal models [122]. Furthermore, recent research proposes p53 DNA repair mechanisms, in which *ATM* plays a role, as potentially being affected in pediatric neurodevelopmental disorders [123].

*GADD45b* up-regulation was detected in some of the datasets of schizophrenia analyzed in our study. *GADD45b* has been proposed as a hub gene differentially expressed in previous bioinformatic studies analyzing microarray datasets from patients with schizophrenia [124]. The overexpression of this GADD45 (growth arrest and DNA-damage-inducible) family member has been found to intensify neuronal loss and cognitive impairments in mice [125], suggesting a possible role of DNA damage response mechanisms in the origin of schizophrenia [126]. Another finding of interest is the up-regulation of *PDHA1* in our data. Anti-PDHA1 antibodies have been found in the sera of patients with schizophrenia, pointing to the mitochondrial dysfunction as a consequence that may underlie the pathogenesis of this condition [127]. Further, *CCDC91* (up-regulated in our analyses) has been proposed as a potential protein biomarker of schizophrenia in a recent proteome-wide association study (PWAS) [128].

*VWA8* and *SNAP29* were significantly up-regulated in our analysis of the bipolar disorder datasets using the CASh method. *VWA8* has been linked to neurological disorders such as autism and bipolar disorder in various genome-wide associated studies [129], while *SNAP29* has been proposed as a candidate of genetically based psychiatric disorders such as schizophrenia and bipolar disorder in previous studies [130,131]. *AQP4*, a well-known drug target for the treatment of bipolar disorder [132], was dysregulated in the analyzed datasets included in our study. Recent studies have reported changes of the protein encoded by *AQP4* in the cerebellum of patients with bipolar disorder, which may provide novel insights into the pathophysiology mechanisms linked to this condition [133].

Regarding the major depression-related genes detected in our study, *EXOSC2* has been proposed as a potential molecular biomarker of major depressive disorder through bioinformatics analysis [134]. To the best of our knowledge, the role of *DPP10* gene in major depression has not been elucidated yet. However, some studies link SNPs in *DPP10* with loneliness and suicidal behaviors, often associated with major depressive disorder [135–137]. Moreover, overexpression of GSTM5 protein was found to be significant in a rat model of depression [138]. Finally, *ZNF184* has been previously proposed as a key gene in the genetic architecture underlying major depressive disorder in GWAS and meta-analysis studies [139,140]. All the mentioned genes were found to be overexpressed in our major depression datasets.

Interestingly, we were able to find some DEGs with a consistent dysregulation across different neuropsychiatric disorders. Peroxiredoxin 6 (Prdx6) is an enzyme encoded by the *PRDX6* gene, which was a DEG detected in the bipolar disorder, schizophrenia and major depression datasets. It is well known that Prdx6 possesses antioxidant activity, with

a role in the maintenance of lipid peroxidation repair, cell metabolism and inflammatory signaling. The altered activity of this enzyme has been associated with central nervous system disorders, and its neuroprotective role of this protein through the inhibition of neuron apoptosis has been reported in animal models [141–143] . Little is known about the role of Prdx6 in ASD and bipolar disorder. However, some studies have confirmed its differential expression in both animal models and human samples of schizophrenia and major depression [144–147] . *SLC4A4* and *Y RNA* were significantly dysregulated in the ASD, schizophrenia and major depression datasets. Recently, it has been described that Slc4a4 is required for normal astrocyte complexity at morphological level and a normal function of the blood-brain barrier [148]. Further, mutations in this gene have been associated with neurological disorders [149,150] and it has been described as a key gene in schizophrenia through bioinformatics strategies [151]. Regarding *Y RNA* gene, it is a small non-coding RNA playing an important role in a range of cellular processes [152]. To the best of our knowledge, there are few reports on the role of these molecules in neurological functions, however some authors have proposed a link between Y RNAs and nervous system disorders [153–155]. Finally, a differential expression of *RAD23B* and *PIAS1* was detected in the ASD, schizophrenia and bipolar disorder datasets. Deficiency of *RAD23B* may affect the normal functioning of motoneurons, which has further implications in the context of Amyotrophic Lateral Sclerosis [156,157]. Regarding *PIAS1*, a protective role against Huntington's Disease and cerebral infarction has been described through the reduction of associated inflammation and apoptosis [158–161], and PIAS genes have been proposed as disease markers in bipolar disorder [162]. However, as far as our knowledge is concerned, the impact of the dysregulation of *RAD23B* and PIAS1 in the neurological disorders analyzed in our work is still to be determined.

Concerning the gene set enrichment analysis of the DEGs identified using the CASh method, we were able to confirm previous findings on the molecular bases of the neurological pathologies studied. For instance, processes related to cardiac muscle cell development in ASD samples are directly linked to vascular abnormalities observed in patients with this phenotype [163]. In Schizophrenia, the regulation of primary metabolic processes and glycine-tRNA ligase activity emerged as significant processes, which are particularly relevant given the metabolic issues associated with this disorder [164]. Similarly, Bipolar Disorder was linked to several key findings in our study, including the positive regulation of lipoprotein lipase activity and synapse and phosphatidylcholine-sterol O-acyltransferase activator activity, which align with the known association of this disorder with altered fatty acids [165]. For Major Depressive Disorder, characterized by inflammation and neurological damage, we identified processes such as "wound healing spreading of cells" and "growth cone" as significant in the context of differential gene expression.

## 5. Conclusions and Limitations

This study highlights the power of Comparative Analysis of Shapley values (CASh) in revealing complex genetic insights into neurological disorders such as Autism Spectrum Disorder (ASD), Schizophrenia, Bipolar Disorder, and Major Depressive Disorder. CASh has been proven as highly effective in identifying differentially expressed genes, many of which are missed by traditional statistical methods, offering a more nuanced understanding of the molecular mechanisms underlying these conditions. These findings open new opportunities for developing innovative diagnostic and therapeutic strategies that may shed light on the etiology of these complex conditions.

However, several limitations should be considered. The inherent complexity of microarray data—such as noise, batch effects, and variability in sample quality—can introduce biases that affect the accuracy of gene expression analysis, despite the rigorous preprocessing and normalization applied. Additionally, the reliance on public datasets may bring biases related to differences in data collection methods, patient selection, and experimental design, potentially limiting the generalizability of our results. To mitigate these issues,

future studies should be conducted to validate the findings by using more diverse cohorts of patients.

Looking ahead, integrating CASh with complementary omics technologies, such as proteomics and metabolomics, promises a more comprehensive view of the pathophysiological processes in brain diseases. This combined approach could significantly improve the development of multi-marker panels, enhancing diagnostic accuracy. Longitudinal studies using CASh could also track disease progression and treatment responses, providing insights into how gene expression evolves over time in relation to disease states.

A further challenge is the computational intensity of CASh, particularly with large datasets. The method requires substantial computational resources, and interpreting Shapley values may be complex. Simplifying the approach—through algorithm optimization or data reduction—would make CASh more accessible for routine clinical and research applications. Additionally, CASh does not account for post-transcriptional modifications or protein-level interactions, which are critical for a complete understanding of disease mechanisms. Future work could address this by integrating CASh with proteomic and metabolomic data to offer deeper insights at the protein level.

Another key limitation is the lack of experimental validation of the identified differentially expressed genes. To confirm the biological relevance of these findings, future studies should incorporate in vitro functional assays, such as gene knockdown or overexpression experiments. Moreover, in vivo studies in animal models would help to further elucidate the roles of these genes in disease mechanisms and assess their potential as therapeutic targets.

Achieving the full potential of CASh will require strong interdisciplinary collaboration. Geneticists, neurologists, oncologists, and bioinformaticians must work together to conduct large-scale studies that validate and refine the gene signatures identified, translating these discoveries into practical clinical applications. By advancing our understanding of the genetic basis of neurological disorders, this research contributes to precision medicine approaches, which ultimately improves patient outcomes and reduces the global burden of these conditions.

**Supplementary Materials:** The following supporting information can be downloaded at: https://www.mdpi.com/article/10.3390/cimb46120812/s1, Figure S1: Exploratory analysis results: Principal Component Analysis (PCA) showing the distribution of gene expression patterns across all the datasets. HPC: hippocampus, PFC: pre-frontal cortex, STR: striatum; Figure S2: Exploratory analysis results: side-by-side volcano plots showing the comparison between the different tests statistics applied to each dataset. HPC: hippocampus, PFC: pre-frontal cortex, STR: striatum; Figure S3: Exploratory analysis results: heatmap showing the distribution of the differentially expressed genes identified by different methods across all the datasets. HPC: hippocampus, PFC: pre-frontal cortex, STR: striatum; Table S1: Technical description of the datasets analyzed in the present study. HPC: hippocampus, PFC: pre-frontal cortex, STR: striatum, BD: bipolar disorder, SCH: schizophrenia, MDD: major depressive disorder; Table S2: Differentially expressed genes obtained for each dataset after statistical analyses. HPC: hippocampus, PFC: pre-frontal cortex, STR: striatum; Table S3: Functional enrichment analysis of the differentially expressed genes obtained in each dataset through the application of Comparative Analysis of Shapley values with raw *p*-values 0.01 and 0.05. HPC: hippocampus, PFC: pre-frontal cortex, STR: striatum, BD: bipolar disorder, SCH: schizophrenia, MDD: major depressive disorder.

**Author Contributions:** Conceptualization, F.J.E. and J.A.C.-M.; methodology, F.J.E., E.V. and J.A.C.-M.; software, F.J.E. and J.A.C.-M.; validation, L.D.-B. and F.J.E.; formal analysis, J.A.C.-M.; investigation, J.A.C.-M., E.V., L.D.-B. and F.J.E.; resources, F.J.E.; data curation, E.V., L.D.-B. and F.J.E.; writing—original draft preparation, J.A.C.-M. and E.V.; writing—review and editing, J.A.C.-M., E.V., L.D.-B. and F.J.E.; visualization, J.A.C.-M.; supervision, E.V., L.D.-B. and F.J.E.; project administration, F.J.E.; funding acquisition, F.J.E. All authors have read and agreed to the published version of the manuscript.

**Funding:** The research group receives funding for research from the University of Jaén (PAIUJA-EI_CTS02_2023) and from the Junta de Andalucía (BIO-302). F.J.E. is partially financed by the Ministry

of Science and Innovation, the State Research Agency (AEI), and the European Regional Development Fund (ERDF—Ref: PID2021-122991NB-C21).

**Institutional Review Board Statement:** Not applicable.

**Informed Consent Statement:** Not applicable.

**Data Availability Statement:** Microarray data were obtained from Gene Expression Omnibus (GEO) database (https://www.ncbi.nlm.nih.gov/geo, accessed on 22 October 2024) as stated above. The custom scripts used for data analysis are deposited in the public repository Zenodo and are available through https://zenodo.org/records/11222132 (accessed on 22 October 2024).

**Conflicts of Interest:** The authors declare no conflicts of interest.

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
