# Peer review of "Enhancing Transcriptomic Insights into Neurological Disorders Through the Comparative Analysis of Shapley Values"

_cimb, doi:10.3390/cimb46120812_

Round 1
Reviewer 1 Report
Comments and Suggestions for Authors
This study by Martinez and colleagues utilized the Comparative Analysis of Shapley values (CASh) on transcriptome data from nine datasets pertaining to neurological illnesses, including autism spectrum disorder (ASD), Schizophrenia (SCH), bipolar disorder (BD), and Major Depressive Disorder (MDD). The method used combines Game Theory with Bootstrap resampling to detect differentially expressed genes (DEGs), providing a reliable alternative to conventional statistical techniques. The proposed method proficiently identified nuanced yet significant molecular patterns, improving the accuracy of transcriptome analysis and offering more understanding of the molecular fingerprints associated with these illnesses. The results may enhance diagnostic methods and facilitate more precise therapy strategies for neurological illnesses. I have the following suggestions and questions related to this paper.
1. In what ways can the Comparative Analysis of Shapley values (CASh) enhance the identification of differentially expressed genes relative to conventional techniques? Please discuss this clearly in the discussion section and what are the limitations of conventional techniques that can be resolved by using CASh.
2. What is the principal differentially expressed genes identified in the research for each neurological disorder? Pleas briefly discuss.
3. In what ways could the insights derived from this work enhance diagnostic methodologies and therapy strategies for neurological disorders? This need to be discussed in the manuscript.
4. What are the constraints associated with employing CASh in transcriptome analysis, and how could forthcoming research mitigate these constraints?
5. Autistic Spectrum Disorder should be Autism Spectrum Disorder. maniac episodes should be manic episodes. encephalon should be brain.
Reviewer 2 Report
Comments and Suggestions for Authors
The study's objective was to analyse gene expression data from nine datasets associated with four heterogeneous neurological pathologies using comparative analysis of shapley values. In contrast to traditional statistical methods, the CASh analysis method, which combines game theory with bootstrap resampling, offers improved detection and interpretation of meaningful gene expression differences and promises to improve the precision of transcriptomic analysis in the future involving complex, multi-genic pathologies.
This study is relevant, covering a new perspective on methods for analyzing transcriptome data, but the major drawback of this work is the relatively weak discussion.
In terms of language quality, the text is well-written, demonstrating a good level of proficiency in English. Although, there are still minor errors, they are easily correctable. The scientific terminology is appropriately used, and the information provided is logically organized.
Title and Abstract
The title of this article is well-written and accurately conveys its goal and content.
The abstract successfully describes the main research problem, objectives, used methods and main results obtained.
Introduction
The introduction outlines the key facts concerning the neurological conditions covered in the article, including information on the aetiology, prevalence, and characteristics of each condition. The introduction's second section discusses microarray data analysis approaches and compares them to standard data analysis techniques and methodologies that use game theory. Drawbacks of classical methods and advantages of CASh transcriptome data analysis are indicated
At the end of the introduction, the main research problem and the research objective are defined.
The manuscript cites current literature published in the in the last five years and provided information is relevant and up to date.
Methods
Chapter 2. “Materials and Methods” contains a detailed description of the used statistical methods, which is systematically organized in subsections, has a high level of detail and provides all necessary justifications for the use of methods to the reader. The first subsection of the chapter describes all the sources used to collect the data for analysis and provides information on the pre-processing and processing of the obtained data. Conventional analyzes for detecting DEGs, which include Welch's t-test with implemented Bayesian-based methods (Empirical Bayes approach), are discussed separately. Also, this chapter contains a subsection that contains information about the Comparative Analysis of Shapley value (CASh) approach. This subsection contains a precise and theoretically based description of the step-by-step statistical methods used in the analysis, defining the equations and applied values, indicating and citing the relevant resources used, such as RStudio packages and other online tools used.
For this portion of the manuscript no improvements are required.
Results
In the first subsection of the results, a detailed description of the dataset used in the study and a comparison of the number of DEGs detected using two distinct strategies for detection - conventional techniques and approach based on CASh method are available. The next part focuses on the functional enrichment analysis of DEGs, which are examined in the datasets of each disease separately. The results of this analysis are visualized in figures (Figure 1 - 5), but they are not properly designed. The text contained in the images is difficult to read due to its small size and I would recommend increasing the font size to make the results easier for the reader to understand and to give them some added value.
Discussion and Conclusions
The discussion in this manuscript is its weakest point. It provides a brief summary of the results acquired without delving deeper into the analysis. For instance, compared to traditional transcriptome analysis, the CASh technique identified more DEGs. But does the increased sensitivity not result in loss of specificity and loss of focus from the most important DEGs? To truly understand the benefits of the CASh method above traditional approaches, answers to these questions are required. Functional enrichment analysis of the DEGs detected using the CASh method gives a very brief impression of the conformity of the obtained results with previous findings in other studies. It would be necessary to rewrite the discussion section entirely.
The conclusions and limitations section identifies research limitations and possible improvements, as well as expresses the main conclusions about the possible application of the CASh method in the future and its potential to improve diagnostic accuracy and other improvements.This section ir well written.
In terms of language quality, the text is well-written, demonstrating a good level of proficiency in English. Although, there are still minor errors, they are easily correctable. The scientific terminology is appropriately used, and the information provided is logically organized.
Reviewer 3 Report
Comments and Suggestions for Authors
I have reviewed the work "Enhancing transcriptomics insights into neurological disorders through the Comparative Analysis of Shapley values," which I find very interesting and valuable in advancing the analysis of neurological disorders. The study demonstrates how, through an innovative strategy using Shapley value analysis, gene expression patterns can be examined effectively. Additionally, the authors acknowledge certain limitations, being aware of the variability inherent in microarray experiments; however, with a solid methodological approach, this method presents a strong alternative.
My only concern is the similarity percentage reported by iThenticate, which is at 49%. Some sections should be revised to address this issue.
Round 2
Reviewer 2 Report
Comments and Suggestions for Authors
The authors of the manuscript have successfully made changes in accordance with the recommendations provided previously. Authors addressed concerns about the potential risk of reduced specificity or loss of focus on the most biologically relevant DEGs using CASh technique. It can also be said that all the figures in the manuscript were successfully redesigned. The most significant improvement is the discussion, which is rewritten with a more specific focus on the topic of the study and includes more current and relevant information.